# Dimethyl fumarate and 4-octyl itaconate are anticoagulants that suppress Tissue Factor in macrophages via inhibition of Type I Interferon

Excessive inflammation-associated coagulation is a feature of infectious diseases, occurring in such conditions as bacterial sepsis and COVID-19. It can lead to disseminated intravascular coagulation, one of the leading causes of mortality worldwide. Recently, type I interferon (IFN) signaling has been shown to be required for tissue factor (TF; gene name *F3*) release from macrophages, a critical initiator of coagulation, providing an important mechanistic link between innate immunity and coagulation. The mechanism of release involves type I IFN-induced caspase-11 which promotes macrophage pyroptosis. Here we find that *F3* is a type I IFN-stimulated gene. Furthermore, *F3* induction by lipopolysaccharide (LPS) is inhibited by the anti-inflammatory agents dimethyl fumarate (DMF) and 4-octyl itaconate (4-OI). Mechanistically, inhibition of *F3* by DMF and 4-OI involves suppression of *Ifnb1* expression. Additionally, they block type I IFN- and caspase-11-mediated macrophage pyroptosis, and subsequent TF release. Thereby, DMF and 4-OI inhibit TF-dependent thrombin generation. In vivo, DMF and 4-OI suppress TF-dependent thrombin generation, pulmonary thromboinflammation, and lethality induced by LPS, *E. coli*, and *S. aureus*, with 4-OI additionally attenuating inflammation-associated coagulation in a model of SARS-CoV-2 infection. Our results identify the clinically approved drug DMF and the pre-clinical tool compound 4-OI as anticoagulants that inhibit TF-mediated coagulopathy via inhibition of the macrophage type I IFN-TF axis.

Inflammation and coagulation are evolutionarily conserved host defence mechanisms that maintain hemostasis by rapidly forming blood clots in response to infection, thereby preventing dissemination of the invading pathogen[1]. Excessive activation of the coagulation cascade is intrinsically linked to increased activation of innate immune pathways and can lead to thrombosis, a pathological deviation from hemostasis[2]. Thrombosis is an integral part of inflammation, and this interplay between innate immunity and coagulation can lead to disseminated intravascular coagulation (DIC), an infection-induced life-

threatening illness characterized by organ dysfunction often accompanied by coagulopathy, and the leading cause of death in intensive care units[3–5].

Tissue factor (TF; also known as coagulation factor 3 (gene name *F3*)) is the primary initiator of trauma-induced coagulation[6]. TF is essential for survival via rapid formation of blood clots. Its deletion is fatal in mice[7–9]. Under normal conditions, TF is expressed at low, basal levels in complex with coagulation factor VII on the membrane of circulating immune cells, such as macrophages, and cells in the blood

✉e-mail: laoneill@tcd.ie

vessel wall[10,11]. This TF:FVIIa complex serves as the key regulator of physiological hemostasis, triggering local coagulation cascade activation which ultimately leads to thrombin generation and blood clot formation. However, under inflammatory conditions such as exposure to lipopolysaccharide (LPS) from gram-negative bacteria, which account for approximately 60% of all sepsis cases[12], *F3* is rapidly induced[13] and its procoagulant activity is increased up to 100-fold[14,15]. Recently, LPS-mediated type I interferon (IFN) and non-canonical inflammasome signaling, which involves caspase-11 or its human homolog caspase-4, have been shown to be critical for TF release from immune cells via pyroptosis, a proinflammatory, lytic form of cell death[16,17]. Extracellular LPS induces caspase-11 via type I IFN production. Cytosolic LPS then binds and activates caspase-11, which in turn cleaves gasdermin D (GSDMD), leading to pyroptosis[18–21]. Procoagulant TF is thereby released on extracellular vesicles through pyroptotic pores in the macrophage cell membrane[22–24] which can lead to aberrant thrombin generation and pathological thrombosis. In addition to its primary function in blood clotting, excessive thrombin generation can trigger detrimental feed-forward inflammation via activation of protease-activated receptors (PARs), which can synergize with Toll-like receptors (TLRs) to induce proinflammatory cytokines and IFN-β[25]. Thrombin can also cleave and activate the proinflammatory cytokine IL-1α[26]. Collectively, this process is termed thromboinflammation[27].

The immunomodulatory agents dimethyl fumarate (DMF), which is clinically approved for the treatment of multiple sclerosis and psoriasis, and the related compound 4-octyl itaconate (4-OI), are potently anti-inflammatory[28–31]. Both are based on endogenous metabolites. DMF is a derivative of the tricarboxylic (TCA) cycle intermediate fumarate, and 4-OI is a derivative of itaconate, which is synthesized in macrophages from the TCA cycle intermediate aconitate via the enzyme ACOD1. 4-OI and itaconate have been shown to have a range of anti-inflammatory effects, including activation of the transcription factor NRF2[29], and inhibition of NLRP3[30], glycolysis[32], TET DNA dioxygenases[33], and the kinase JAK1[34]. Some of these effects such as NRF2 activation[35] and inhibition of glycolysis[28], NLRP3[31], and JAK1[36] are shared with DMF. Both DMF and 4-OI have also been shown to inhibit IFN-β production[29,37].

We have therefore examined whether DMF and 4-OI might block inflammation-associated coagulation via inhibition of type I IFN signaling and the release of TF. We found that *F3* is a type I IFN-stimulated gene (ISG), and inhibition of *F3* induction by DMF and 4-OI is likely to involve suppression of *Ifnb1*. DMF and 4-OI also block type I IFN- and caspase-11-driven macrophage pyroptosis, thereby inhibiting the release of procoagulant TF, leading to suppression of inflammation-associated coagulation in vivo in response to LPS, *E. coli*, *S. aureus*, and SARS-CoV-2. DMF and 4-OI are therefore identified as anticoagulants that inhibit the macrophage type I IFN-TF axis with potential to limit infection-associated coagulopathy.

## Results

### DMF and 4-OI inhibit LPS-mediated *F3* induction and TF-dependent thrombin generation in macrophages

Recent evidence has identified macrophages as important contributors to the total pool of TF available for the initiation of blood clotting[6,22]. Using publicly available data[38], we first confirmed that macrophages are a major expressor of *F3* mRNA (Supplementary Fig. 1), supporting the use of macrophages as an appropriate cell type for this study. Differential gene expression of LPS-stimulated mouse bone marrow-derived macrophages (BMDMs) revealed that DMF suppressed induction of *F3* in addition to *F7* (Fig. 1a), suggesting that DMF may inhibit TF:FVII(a) complex formation. DMF also inhibited induction of *F10* and *F2r* (PAR1) which indicated that DMF may be a potent inhibitor of the extrinsic pathway of coagulation as well as downstream thrombin signaling. Furthermore, expression of *Casp4/11* and *Gsdmd*, which are required for non-canonical inflammasome-

mediated pyroptosis, were also reduced (Fig. 1a). We, therefore, confirmed by PCR that DMF and 4-OI downregulated LPS-induced mRNA expression of *F3* in both BMDMs and human peripheral blood mononuclear cells (PBMCs) (Fig. 1b, c), in addition to suppressing TF protein levels in BMDMs (Fig. 1d, e). We next assessed the effect of DMF and 4-OI on TF-dependent thrombin generation, a functional readout of TF procoagulant activity. Thrombin generation was assessed using FXII-deficient plasma to examine the effect of DMF and 4-OI on TF-dependent, and therefore FXII-independent, thrombin generation[39]. After stimulating BMDMs with LPS, DMF and 4-OI both significantly increased TF-dependent thrombin generation lagtime in normal platelet-poor plasma (Fig. 1f–h), indicating functional inhibition of thrombin generation via suppression of *F3*.

### *F3* is a type I IFN-stimulated gene with JAK-STAT-dependency

To assess if DMF and 4-OI regulate type I IFN in the context of coagulation, we first examined their effect on LPS-induced type I IFN production. Consistent with previous studies[29,37], DMF and 4-OI blocked LPS-mediated transcriptional induction of *Ifnb1* (Fig. 2a) and subsequent IFN-β release from BMDMs (Fig. 2b). We next hypothesized that *F3* is induced directly via type I IFN signaling, and found that LPS-mediated induction of *F3* is significantly decreased in BMDMs from *Ifnar*[−/−] mice (Fig. 2c), which do not express the type I IFN receptor. Furthermore, IFN-β itself induced *F3* after 4 h, which was transient (Fig. 2d), indicating that *F3* is an ISG. IFN-β-induced *F3* was inhibited by DMF and 4-OI (Fig. 2e).

Both DMF and 4-OI exert their anti-inflammatory effects in part via NRF2 activation[29,35], and NRF2 activation has recently been shown to inhibit type I IFN signaling[40]. We confirmed that DMF and 4-OI drive the NRF2-dependent genes *Gclm* and *Hmox1* (Supplementary Fig. 2a, b). However, knockdown of NRF2 in BMDMs did not alter the suppression of LPS-induced *F3* by DMF and 4-OI (Supplementary Fig. 2c). We confirmed the knockdown in Supplementary Fig. 2d, whereby *Gclm* expression was decreased in the NRF2-deficient cells. This suggests that the mechanism of inhibition of *F3* by DMF and 4-OI is NRF2-independent.

Type I IFNs act upon their receptor, IFNAR, to induce hundreds of ISGs via the JAK-STAT signaling pathway[41]. We next explored this component of the type I IFN pathway further and found that LPS-induced *F3* was blocked by baricitinib (Fig. 2f), a specific JAK1/2 inhibitor that limits type I IFN signaling. It has recently been shown that JAK1 is modified and inhibited by both DMF and 4-OI[34,36]. Knockdown of JAK1 significantly reduced LPS-mediated *F3* induction (Fig. 2g; knockdown confirmed in Supplementary Fig. 3a), further indicating a JAK1-dependency for *F3* induction. We next analyzed transcription factor binding sites in the region of the *F3* gene promoter using the publicly available Interferome database[42] which predicted STAT-binding sites in the promoter of the *F3* gene (Fig. 2h; coordinates in Supplementary Table 1). We verified these findings using the ChIP-Atlas enrichment analysis tool[43] which also predicted STAT- and IRF-binding sites in the downstream region of the human *F3* promoter (Fig. 2i), further suggesting a type I IFN- and STAT-dependency for *F3*. The inhibition by DMF and 4-OI of *F3* induction in response to LPS would therefore appear to involve suppression of type I IFN production and signaling via JAK-STATs.

### DMF and 4-OI inhibit caspase-11-mediated pyroptosis, suppressing TF release from pyroptotic macrophages and subsequent thrombin generation

We next examined whether DMF and 4-OI might block the release of TF by inhibiting type I IFN- and caspase-11-mediated pyroptosis. As caspase-11 is an ISG[44] (as confirmed in Supplementary Fig. 3b), DMF and 4-OI dose-dependently blocked LPS-induced *Casp11* expression in BMDMs (Fig. 2j; Supplementary Fig. 3c, d). DMF and 4-OI also inhibited induction of the ISGs *Isg15* and *Usp18* (Supplementary Fig. 3e, f),

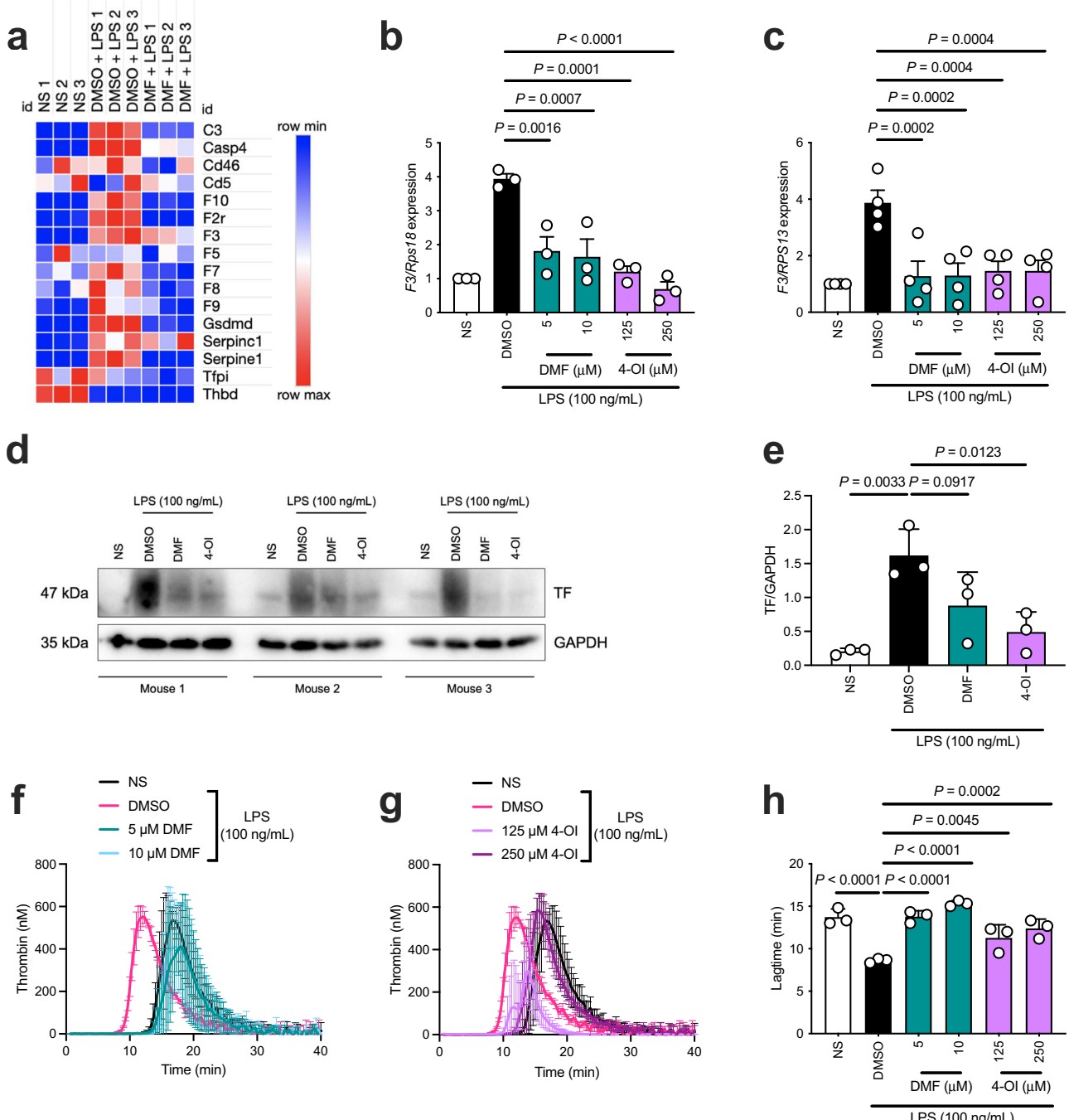

**Fig. 1 | DMF and 4-OI inhibit LPS-mediated *F3* induction and TF-dependent thrombin generation in macrophages. a** Heatmap from RNA sequencing of selected inflammation-associated coagulation genes in LPS-stimulated mouse macrophages (BMDMs) pre-treated (3 h) with DMF compared to DMSO control prior to LPS stimulation (4 h). **b** BMDMs (*n* = 3) and **c** PBMCs (*n* = 4) were pre-treated with DMSO, DMF, or 4-OI (1 h) prior to LPS stimulation (3 h) and harvesting cell lysates. Quantification of *F3* mRNA by qRT-PCR in **b** BMDMs (DMSO–250 μM 4-OI, *P* = 0000178) and **c** PBMCs. **d** Western blot and **e** densitometry analysis of TF in BMDM cell lysates pre-treated with DMSO, DMF, or 4-OI (1 h) prior to LPS stimulation (3 h), with GAPDH as loading control. **f–h** BMDMs were pre-treated with DMSO, **f** DMF, or **g** 4-OI (1 h) before LPS priming (3 h). Thrombin generation was measured in BMDMs in situ in plate wells using FXII-deficient plasma. **h** Lagtime represents time-to-clot formation. NS–DMSO, *P* = 0.000015; DMSO–5 μM DMF, *P* = 0.00001; DMSO–10 μM DMF, *P* = 0.0000006. Data from **b**, **c** are mean ± SEM from 3, 4 independent experiments. Blot from **d** is three mice from three independent experiments. Data from **e–h** are mean ± SD from three independent experiments. *P* values calculated one-way ANOVA for multiple comparisons. Source data are provided as a Source Data file.

indicating a broad reduction in type I IFN signaling. Detection of cytosolic LPS activates caspase-11, which in turn cleaves GSDMD to form pyroptotic pores[20,21]. We therefore next assessed caspase-11-mediated pyroptosis. BMDMs were primed with LPS and subsequently transfected with LPS (which delivers LPS to the cytosol to activate caspase-11[45]). Pre-treatment, but not post-treatment, of BMDMs

(Supplementary Fig. 3g) with DMF and 4-OI dose-dependently blocked pyroptosis, as measured by LDH release (Fig. 2k) and PI staining (Supplementary Fig. 3h), consistent with induction of *Ifnb1* and subsequent *Casp11* as being the process being targeted here. Caspase-11- and pyroptosis-mediated TF release (assessed by release into the supernatant) was also suppressed by DMF and 4-OI (Fig. 2l–n),

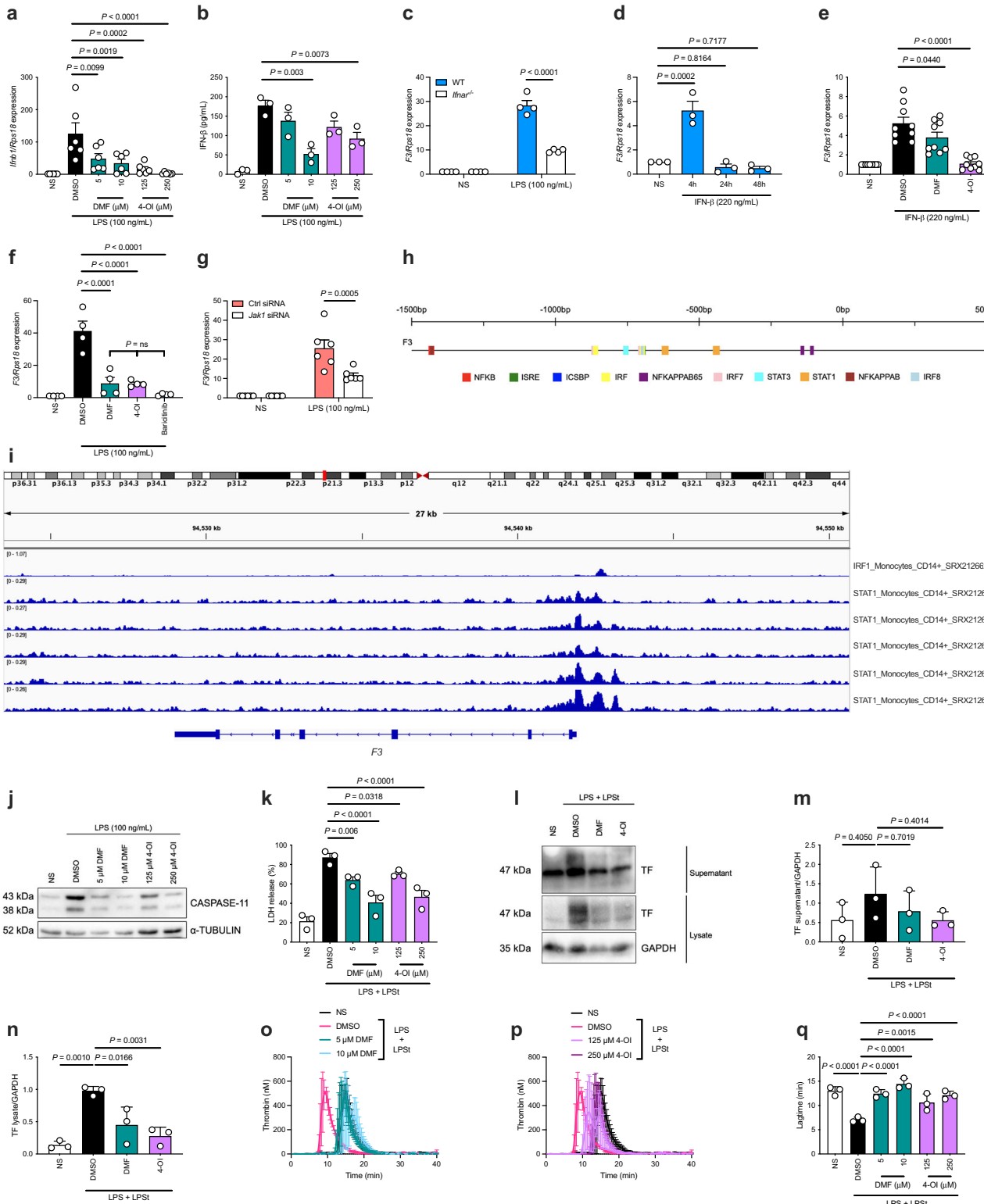

resulting in decreased TF procoagulant activity and TF-dependent thrombin generation after activation of the non-canonical inflammasome, shown in Fig. 2o for DMF, Fig. 2p for 4-OI, and lagtime in Fig. 2q.

**Therapeutic treatment with DMF and 4-OI suppresses LPS-induced thrombin generation in vivo, improving survival in mice**

Following our findings that DMF and 4-OI block the induction and release of TF in response to LPS, thereby blocking thrombin generation in vitro, we next investigated the effects of DMF and 4-OI in vivo in a systemic model of LPS-induced inflammation and coagulopathy. Intraperitoneal LPS injection significantly induced thrombin generation in mice (Fig. 3a–f). Therapeutic administration (2 h *after* LPS injection) revealed that DMF (Fig. 3a) and 4-OI (Fig. 3b) significantly inhibited thrombin generation in vivo, as did baricitinib (Fig. 3c) and the specific anti-TF antibody, 1H1, which blocks TF:FVIIa assembly[46] (Fig. 3d). These data are quantified in Fig. 3e, f. Collectively, this demonstrates the multiple points of TF control as the compounds used target three separate stages of TF activity: *F3* induction and

**Fig. 2 | *F3* is a type I IFN-stimulated gene with JAK-STAT-dependency, and DMF and 4-OI suppress TF release from macrophages via inhibition of type I IFN- and caspase-11-mediated pyroptosis. a** BMDMs were pre-treated with DMSO, DMF, or 4-OI (3 h) before LPS priming (16 h) and harvesting cell lysates (*n* = 6). *Ifnb1* mRNA was quantified by qRT-PCR. DMSO–250 μM 4-OI, *P* = 0.00004. **b** BMDMs were pre-treated with DMSO, DMF, or 4-OI (3 h) prior to LPS priming (6 h). IFN-β in the supernatant was measured by enzyme-linked immunosorbent assay (ELISA) (*n* = 3). **c** BMDMs from WT and *Ifnar⁻/⁻* mice (*n* = 4) were stimulated with LPS (3 h). *F3* mRNA was quantified by qRT-PCR. LPS 3 h, WT-KO, *P* = 0.0000000488. **d** BMDMs were stimulated with recombinant mouse IFN-β for a timecourse as indicated (*n* = 3). *F3* mRNA was quantified by qRT-PCR. **e** BMDMs pre-treated with DMSO, DMF, or 4-OI (1 h) before stimulation with recombinant mouse IFN-β (4 h) and harvesting cell lysates (*n* = 9). *F3* mRNA was quantified by qRT-PCR. DMSO–4-OI, *P* = 0.00000018. **f** BMDMs (*n* = 4) were pre-treated with DMSO, DMF, 4-OI, or baricitinib (1 h) before LPS stimulation (3 h). *F3* mRNA was quantified by qRT-PCR. DMSO–DMF, *P* = 0.0000144; DMSO–4-OI, *P* = 0.0000133; DMSO–baricitinib, *P* = 0.00000134. **g** BMDMs (*n* = 6) were transfected with Ctrl or *Jak1* siRNA and stimulated with LPS (3 h). *F3* mRNA was quantified by qRT-PCR. **h** Predicted transcription factor sites in the promoter of the *F3* gene via the Interferome database[42]. Data presented shows the predicted location of transcription factors from the region spanning −1500 bp to +500 bp from the start site of the *F3* gene promoter. **i** The location of STAT1- and IRF1-transcription factor sites in the promoter of the human *F3* gene, analyzed from enrichment analysis of 6 independent experiments from publicly available data using the ChIP-Atlas[43]. **j** Representative western blot of CASPASE-11 in BMDM cell lysates pre-treated with DMSO, DMF, or 4-OI (1 h) prior to LPS stimulation (3 h), with α-TUBULIN as loading control. Blot is representative of three independent experiments. **k** BMDMs (*n* = 3) were pre-treated with DMSO, DMF, or 4-OI (1 h) before priming with LPS (3 h) and LPS transfection (16 h). Pyroptosis is represented as percentage cell death measured by LDH release in BMDM supernatants. DMSO–10 μM DMF, *P* = 0.0000247; DMSO–250 μM 4-OI, *P* = 0.000084. **l** Representative western blot of TF in BMDMs pre-treated with DMSO, DMF, or 4-OI (1 h) prior to LPS priming (3 h) and LPS transfection (16 h). Blots are representative of three independent experiments. Densitometry analysis of TF in BMDM (**m**) supernatants and **n** cell lysates, with GAPDH as loading control. **o–q** BMDMs were pre-treated with DMSO, **o** DMF, or **p** 4-OI (1 h) before priming with LPS (3 h) and LPS transfection (16 h). Thrombin generation was measured in BMDMs in situ in plate wells using FXII-deficient plasma. **q** Lagtime represents time-to-clot formation. NS–DMSO, *P* = 0.000017; DMSO–5 μM DMF, *P* = 0.000044; DMSO–10 μM DMF, *P* = 0.0000019; DMSO–250 μM 4-OI, *P* = 0.00007985. Data from **a–g**, **k** are mean ± SEM from 3–6 independent experiments. Data from **m–q** are mean ± SD from 3 independent experiments. *P* values calculated using two-tailed Student's *t* test for paired comparisons or one-way ANOVA for multiple comparisons. Source data are provided as a Source Data file.

suppression of pyroptosis via inhibition of type I IFN (DMF and 4-OI), downstream JAK1 activity (baricitinib), and extrinsic-mediated coagulation via inhibition of TF:FVIIa (1H1).

We next assessed the effects of DMF and 4-OI in a model of LPS-induced lethality. We challenged mice intraperitoneally with a lethal dose of LPS for 24 h, before treating therapeutically with DMF or 4-OI. We also included a mouse group that received heparin as a control anticoagulant and caspase-11 inhibitor[47]. All DMF- and heparin-treated mice survived a lethal dose of LPS, with 4-OI reducing mortality by 80% compared with PBS-treated mice (Fig. 3g). Clinical scores of mice (assessing weight loss, activity level, eye closure, appearance of fur and posture) highlighted the protective roles of DMF and 4-OI when administered 24 h after LPS compared with PBS-treated mice (Fig. 3h). This is highlighted by the divergence in clinical scores of DMF- and 4-OI-treated mice compared with PBS-treated mice when the treatments were administered 24 h after LPS injection (Fig. 3h). Prophylactic administration of DMF and 4-OI also improved survival, with all DMF- and 4-OI-treated mice surviving a lethal dose of LPS, as did all *Casp11⁻/⁻* mice (Supplementary Fig. 4a), thus supporting the critical role played by caspase-11 in LPS-induced sepsis and coagulopathy. Clinical scoring of mice further showed the protective effects of DMF and 4-OI (Supplementary Fig. 4b). Taken together, these results indicate that DMF and 4-OI can protect against LPS-induced lethality, with inhibition of type I IFN- and TF-mediated thrombin generation likely to be a critical aspect of their protective effects.

### DMF and 4-OI inhibit TF release and TF-mediated thrombin generation following infection with *E. coli* and *S. aureus*

Having shown an inhibitory effect against LPS, we next turned to infectious agents. *Escherichia coli* (*E. coli*) infection represents a clinically relevant model of TLR4, gram-negative bacterial-mediated coagulation. Consistent with our LPS data, DMF significantly inhibited, with 4-OI non-significantly reducing, *E. coli*-induced thrombin generation in vivo (Fig. 3i-k).

We next assessed whether the anticoagulant effects of DMF and 4-OI could extend beyond gram-negative bacterial infection. We thus tested if DMF and 4-OI could inhibit coagulopathy induced by two different strains of the gram-positive bacterium *Staphylococcus aureus* (*S. aureus*), which is driven by *F3* induction and TF procoagulant activity[48]. We first demonstrated inhibition of *F3* induction by DMF and 4-OI in vitro when BMDMs were infected with *S. aureus* (Fig. 3l). In vivo, DMF and 4-OI both reduced thrombin generation following intraperitoneal injection with *S. aureus* (Fig. 3m, n). Consistent with this

reduction in TF-dependent thrombin generation, DMF and 4-OI markedly reduced TF deposition in the lungs of intraperitoneally-infected mice (Fig. 3o). DMF and 4-OI also reduced thrombin generation following intravenous injection of *S. aureus* (Supplementary Fig. 5a–c). Histological analysis showed decreased TF expression in the lungs of DMF- and 4-OI-treated mice in this model compared with PBS-treated mice (Supplementary Fig. 5d), demonstrating inhibition of *S. aureus*-driven coagulopathy via different modes of infection. As *S. aureus*-mediated coagulopathy occurs in a caspase-11-independent manner[49], this finding highlights the importance of the inhibition by DMF and 4-OI on *F3* induction and subsequent TF procoagulant activity in blocking aberrant thrombin generation in bacterial-driven sepsis.

### 4-OI suppresses lung inflammation with associated coagulopathy in mice after SARS-CoV-2 infection

Having established that DMF and 4-OI potently inhibit type I IFN- and TF-mediated thrombin generation in vivo following bacterial infection, we next assessed the broader anticoagulant effects of these compounds in a model of viral-induced coagulopathy. TF is a key contributor to viral-associated coagulation[50], so we next tested the effects of DMF and 4-OI on the prothrombotic effect of the viral dsRNA mimetic, poly(I:C). Both DMF and 4-OI inhibited *F3* induction in BMDMs in vitro after stimulation with poly(I:C) (Fig. 4a). Consistent with our LPS data, poly(I:C) stimulation also induced *Casp11*, which was potently inhibited by DMF and 4-OI (Fig. 4b).

Coagulopathy is a hallmark of COVID-19-associated pathology, and dysregulated immunothrombosis is a critical driver of this aberrant state of coagulation[51–54]. Inflammasome activation[55] (in particular via caspase-11 activation[56]) and TF expression[57,58] and procoagulant activity[59,60] have been shown to exacerbate SARS-CoV-2-associated immunothrombosis and coagulopathy. Recent evidence indicates a prothrombotic shift in the phenotype of monocytes following COVID-19 infection[61]. Furthermore, dysregulated type I IFN signaling is a critical mediator of COVID-19-associated pathology[62–64]. We therefore tested 4-OI in a mouse model of SARS-CoV-2 infection and assessed broader anticoagulant effects. SARS-CoV-2 infection in mice induced extensive lung inflammation and an increase in the presence of inflammatory cells in the parenchyma (Fig. 4c; lower left-hand quadrant). 4-OI treatment reduced lung inflammation and infiltration of inflammatory cells (Fig. 4c; lower right-hand quadrant). Diffuse pulmonary damage and coagulopathy following SARS-CoV-2 infection is characterized by elevated lung von Willebrand Factor (vWF)

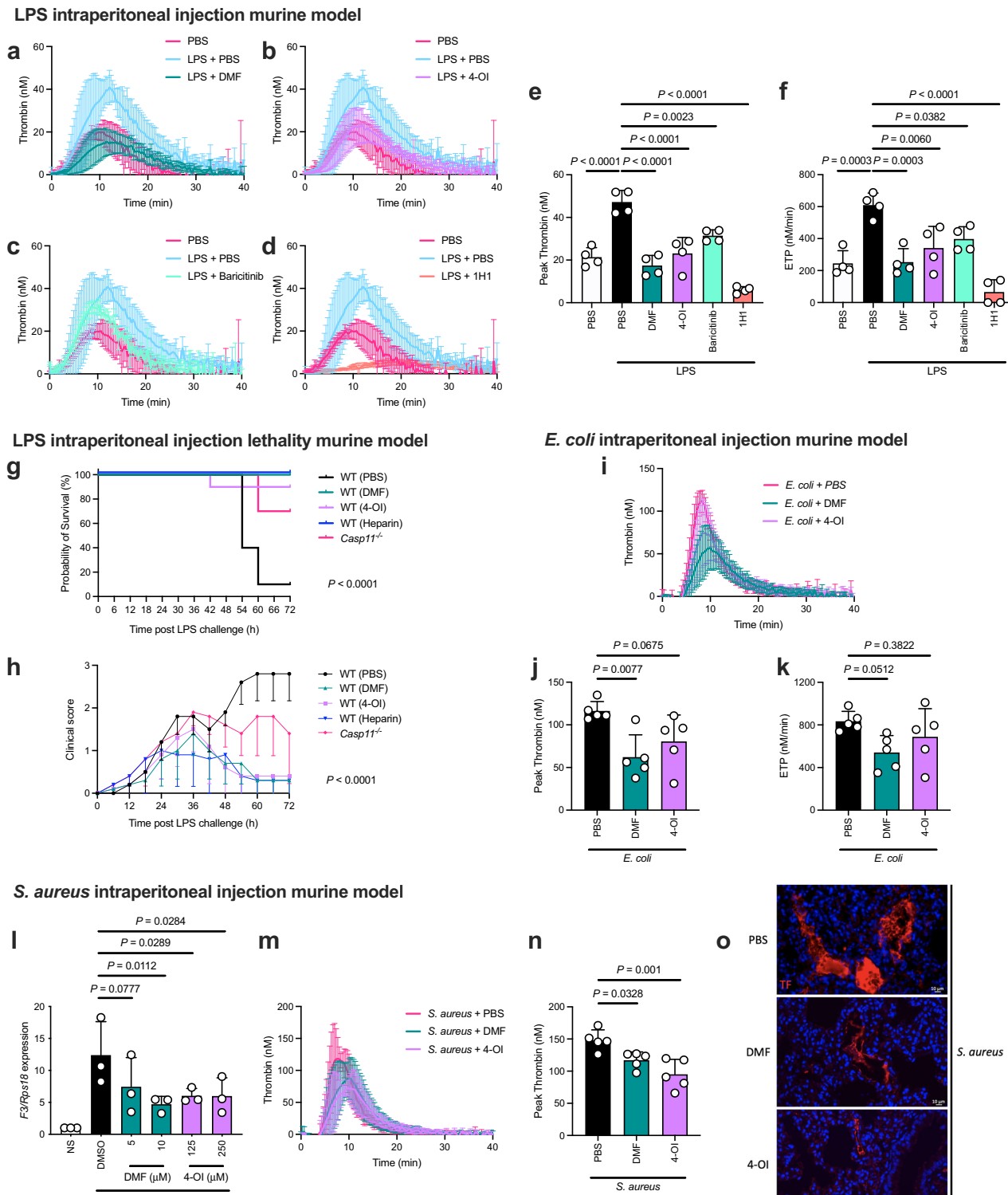

**LPS intraperitoneal injection murine model**

**LPS intraperitoneal injection lethality murine model**

***E. coli* intraperitoneal injection murine model**

***S. aureus* intraperitoneal injection murine model**

expression[65] and extensive intraalveolar fibrin deposition[51]. Following SARS-CoV-2 infection, we found a significant elevation of vWF and fibrinogen/fibrin surrounding the parenchyma, blood vessels, and airways in the lungs (Fig. 4d–g; lower left-hand quadrant). Mice that received 4-OI had significantly lower levels of both vWF and fibrinogen/fibrin (Fig. 4d–g; lower right-hand quadrant), which was indistinguishable from mock-infected mice. Increased collagen deposition indicates dysregulated tissue repair in the lungs, and SARS-CoV-2-infected mice also displayed elevated collagen deposition (Fig. 4, h; lower left-hand quadrant), a marker of lung damage induced by

inflammation-associated coagulation[66]. There was a significant reduction in collagen deposition in the lungs of 4-OI-treated mice (Fig. 4h, i; lower right-hand quadrant).

4-OI treatment also reduced total numbers of neutrophils (Fig. 4j), lymphocytes (Fig. 4k), and leukocytes (Fig. 4l) in the bronchoalveolar lavage fluid (BALF) of SARS-CoV-2-infected mice. This is notable as neutrophils are a significant contributor to SARS-CoV-2-associated inflammation-driven coagulopathy[67,68]. 4-OI also significantly reduced infectious viral load in the BALF (Fig. 4m) and lung homogenates (Fig. 4n) of infected mice, indicating an anti-viral effect of 4-OI,

**Fig. 3 | Therapeutic treatment with DMF and 4-OI suppresses thrombin generation in vivo induced by LPS, *E. coli*, and *S. aureus*, improving survival in mice. a–f** Mice were intraperitoneally injected with 1 mg/kg LPS (2 h) followed by treatment with PBS, **a** 50 mg/kg DMF, **b** 50 mg/kg 4-OI, **c** 50 mg/kg baricitinib, or **d** 15 mg/kg 1H1 anti-TF antibody (for a further 4 h) via intraperitoneal injection. Citrated plasma was harvested and thrombin generation was assessed (*n* = 4 per group). Thrombin generation in **a–d** is compared relative to PBS-treated mice ±LPS challenge. **e** Peak thrombin in mouse citrated plasma treated as in **a–d**. PBS−LPS + PBS, *P* = 0.00000637; LPS + PBS−LPS + DMF, *P* = 0.0000008; LPS + PBS−LPS + 4-OI, *P* = 0.0000167; LPS + PBS−LPS + 1H1, *P* = 0.0000000055. **f** Total thrombin generation (ETP = endogenous thrombin potential) in mouse citrated plasma treated as in **a–d**. LPS + PBS−LPS + 1H1, *P* = 0.00000135. **g** Kaplan–Meier survival curve of mice injected intraperitoneally with 15 mg/kg LPS (24 h) followed by treatment with PBS, 50 mg/kg DMF, 50 mg/kg 4-OI, or 200 IU/kg heparin (for a further 48 h) (*n* = 10 per group). LPS + PBS−treatment groups, *P* = 0.0000003. **h** Mice from **g** were scored clinically (assessing weight loss, activity level, eye closure, and appearance of fur and posture) every 6 h for the duration of the 72 h experiment. LPS + PBS−LPS + DMF, *P* = 0.00000001997. Data points indicate individual mice in **g**, **h**. For **g**, **h** Mantel–Cox survival analysis was performed.

**i** Mice were intraperitoneally injected with 1 × 10⁷ CFU *E. coli* (CFT073) and co-treated with PBS, 50 mg/kg DMF, or 50 mg/kg 4-OI (6 h), followed by supplemental treatment via intraperitoneal injection with PBS, 50 mg/kg DMF, or 50 mg/kg 4-OI (for a further 18 h). Citrated plasma was harvested and thrombin generation was assessed (*n* = 5 per group). **j** Peak and **k** total thrombin generation in mouse citrated plasma treated as in **i** (*n* = 5 per group). **l** BMDMs were pre-treated with DMSO, DMF, or 4-OI (1 h) before infection with 5 × 10⁶ CFU *S. aureus* (USA300-LAC) (3 h) and harvesting cell lysates. *F3* mRNA was quantified by qRT-PCR. **m** Mice were intraperitoneally injected with 5 × 10⁸ CFU *S. aureus* (PS80) and co-treated with PBS, 50 mg/kg DMF, or 50 mg/kg 4-OI (6 h), followed by supplemental treatment via intraperitoneal injection with PBS, 50 mg/kg DMF, or 50 mg/kg 4-OI (for a further 18 h). Citrated plasma was harvested and thrombin generation was assessed (*n* = 5 per group). **n** Peak thrombin generation in mouse citrated plasma treated as in **m**. **o** TF-positive regions (red) in the lungs of mice treated as in **m**. Representative lung tissue sections are shown. Magnification ×20. Scale bar = 10 µm. Data from **a–k** and **m**, **n** are mean ± SD. Data from **l** are mean ± SEM from three independent in vitro experiments. *P* values were calculated using a two-tailed Student's *t* test for paired comparisons or one-way ANOVA for multiple comparisons. Source data are provided as a Source Data file.

consistent with previous reports[69,70]. Whilst SARS-CoV-2-infected mice exhibited significant weight loss, the weight loss of 4-OI-treated mice following SARS-CoV-2 infection was not significantly different compared with mock-infected mice (Supplementary Fig. 6a). Clinical scoring of mice (assessing weight loss, activity level, eye closure, appearance of fur and posture) highlighted the protective role of 4-OI in this model (Fig. 4o). At 6 days post infection, *Ifnb1* expression was significantly attenuated by 4-OI (Fig. 4p), providing further indication that suppressing type I IFN signaling is critical in reducing TF-mediated coagulopathy in COVID-19. Notably, 4-OI significantly increased *Nfe2l2* expression after SARS-CoV-2 infection (Fig. 4q). Although NRF2 knockdown did not directly alter LPS-induced *F3* expression (previously shown in Supplementary Fig. 2c), NRF2 activation by 4-OI may contribute to decreased type I IFN signaling[40,69] by an as-yet-unknown mechanism and therefore decrease coagulopathy in COVID-19. Furthermore, 4-OI limited SARS-CoV-2-induced expression of the proinflammatory genes *Ifng* and *Il1b* (Supplementary Fig. 6a, b), indicating a broad reduction in inflammation upon 4-OI treatment. Finally, *F3* expression was elevated in PBMCs from SARS-CoV-2-infected patients compared with healthy controls (Fig. 4r), in addition to *CASP4* (Fig. 4s), confirming the procoagulant genotype associated with COVID-19.

## Discussion

TF mediates hemostasis upon vessel injury, but infection and other proinflammatory stimuli can trigger rapid *F3* induction in macrophages to provoke aberrant coagulation, with inflammation-associated coagulation an underlying driver of sepsis-associated coagulopathies. In particular, it has recently been shown that type I IFN signaling is a critical mediator of coagulation[17]. Type I IFN induction facilitates the increased expression of caspase-11, which is activated and cleaved upon detection of cytosolic LPS, leading to the formation of pyroptotic pores in macrophages[18–21]. Procoagulant TF is released from macrophages on extracellular vesicles through these pyroptotic pores, initiating rapid, pathological blood clotting[22–24].

Whilst current anticoagulant therapies are effective, they are associated with an increased bleeding risk for patients[27,71]. This bleeding is significantly enhanced in septic patients and can manifest as life-threatening bleeds such as hemorrhagic stroke[72]. Thus, developing anticoagulants without an associated bleeding risk is a key goal for the treatment of coagulopathies[73]. Notably, no specific TF inhibitor has been clinically approved for anticoagulation in humans due to the essential role of TF in maintaining hemostasis and the associated bleeding risk of solely targeting the TF:FVIIa complex. Furthermore, it was recently shown that the anticoagulant heparin does not improve the likelihood of survival-to-hospital-discharge when administered to critically ill COVID-19 patients[74], highlighting the urgent need to counteract the innate immune signaling component of coagulopathies and develop broad-spectrum treatments for coagulopathies associated with excessive inflammation.

This recognition of the role of innate immune signaling as a key contributor to pathological thrombosis has led to a recent surge in assessing clinically approved anti-inflammatory therapies as potential anticoagulants[75–78]. The clinically approved drug DMF and the preclinical tool compound 4-OI are potently anti-inflammatory agents and share multiple targets, including NRF2 activation[29,35], and inhibition of glycolysis[28,32] and NLRP3 inflammasome activation[30,31]. Furthermore, they also inhibit type I IFN induction[29,37] which contributes to aberrant coagulation. Therefore, we hypothesized that DMF and 4-OI might regulate coagulation via modulation of the type I IFN-TF axis in macrophages.

In our study, we have identified DMF and 4-OI as inhibitors of inflammation-associated coagulation and report for the first time their anticoagulant effects via suppression of the type I IFN-TF axis in macrophages (Supplementary Fig. 7), which has been implicated in aberrant coagulation[17]. Mechanistically, we have found that *F3* is an ISG with predicted STAT-binding sites in the *F3* gene promoter, and that DMF and 4-OI likely inhibit *F3* induction at least in part via suppression of *Ifnb1*. Additionally, they block caspase-11-mediated release of TF via suppression of macrophage pyroptosis, again most likely as a consequence of inhibiting type I IFN production with a subsequent decrease in caspase-11 expression. DMF and 4-OI protected against aberrant TF-mediated thrombin generation in vivo in models of bacterial- (both gram-negative and gram-positive) and viral infection and via multiple routes of administration. In particular, our results showing reduced deposition of TF in the lungs with DMF and 4-OI treatment might be particularly relevant in acute respiratory distress syndrome, a leading cause of death in bacterial sepsis due to wholesale destruction of the lung endothelial barrier induced by excessive inflammation and coagulation[79]. 4-OI also reduced deposition of fibrinogen/fibrin in the lungs of mice following SARS-CoV-2 infection. This is likely due to inhibition of excessive TF-dependent thrombin generation, which cleaves fibrinogen into fibrin. The presence of persistent fibrin amyloid microclots has been linked with the long-term debilitating effects of long COVID[80], and furthermore, elevated type I IFN production has been detected at least 8 months after the onset of COVID-19 infection[81]. This suggests potential utility for 4-OI and inhibitors of the type I IFN-TF axis in long COVID. Furthermore, vWF deposition, which was blocked by 4-OI, has recently been shown to activate proinflammatory signaling in macrophages[82]. This suggests that, in addition to its primary anti-inflammatory and anticoagulant properties, 4-OI might also inhibit downstream amplification of thromboinflammation.

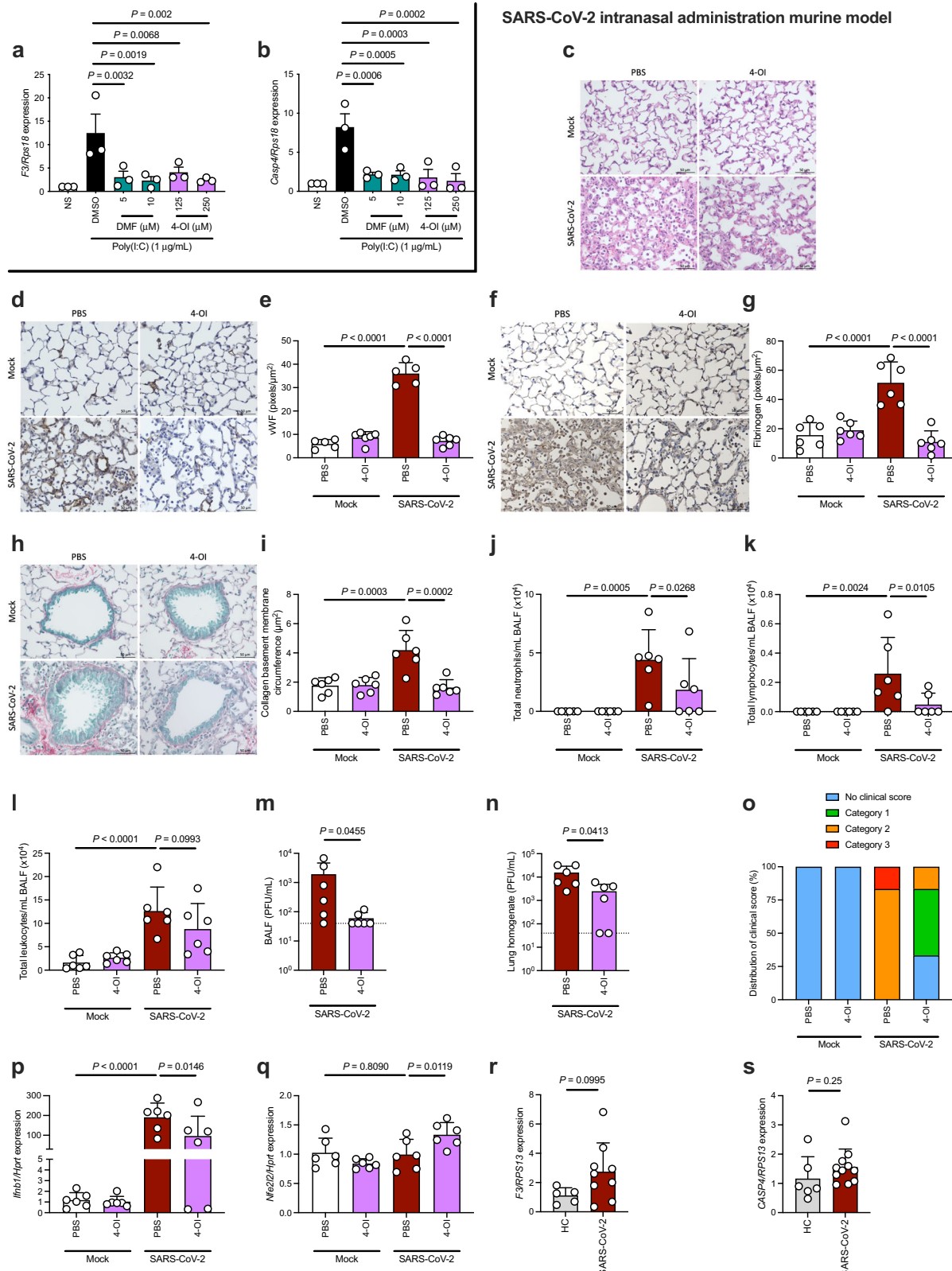

We also found that 4-OI reduces viral titers following SARS-CoV-2 infection. Emerging evidence points to itaconate derivatives including 4-OI as being anti-viral. For example, 4-OI attenuates pathology associated with HSV-1 and- 2, vaccinia, and zika viruses[69] as well as influenza A[70]. It has recently been shown that thrombin can directly cleave the SARS-CoV-2 spike protein, augmenting viral entry into the lungs[83]. This suggests a broader anti-viral effect of

therapeutic thrombin inhibition in the treatment of COVID-19, as this will likely suppress pathological type I IFN production (as a result of reduced viral uptake) as well as thromboinflammation. This also might be a critical downstream effect of the inhibition of TF-mediated thrombin generation by 4-OI following SARS-CoV-2 infection and may explain the reduction in viral titers following 4-OI treatment.

**Fig. 4 | 4-OI suppresses lung inflammation with associated coagulopathy after SARS-CoV-2 infection. a, b** BMDMs were pre-treated with DMSO, DMF, or 4-OI (1 h) before poly(I:C) stimulation (3 h) and harvesting cell lysates. **a** *F3* and **b** *Casp4* (caspase-11) mRNA was quantified by qRT-PCR. **c** Representative images of H&E staining examining histological changes in lung structure and inflammatory cell infiltration in the primary airways, parenchyma, and vasculature from male K18-hACE2 mice infected with $10^3$ PFU SARS-CoV-2 (Wuhan isolate; VIC01/2020) or mock infection and intranasally treated with PBS or 4-OI (10 mg/kg) from 1 day post infection, with daily treatments continuing until the termination of the experiment at day 6 post infection (*n* = 5–6 per group). **d** Representative images and **e** quantification of vWF staining in lung sections from mice treated as in **c** (*n* = 5–6 per group). Mock+PBS–SARS-CoV-2 + PBS, *P* = 0.00000000002; SARS-CoV-2 + PBS–SARS-CoV-2 + 4-OI, *P* = 0.00000000002. **f** Representative images and **g** quantification of fibrinogen staining in lung sections from mice treated as in **c** (*n* = 5, 6 per group). Mock+PBS–SARS-CoV-2 + PBS, *P* = 0.000016; SARS-CoV-2 + PBS–SARS-CoV-2 + 4-OI, *P* = 0.000002887. **h** Representative images and **i** quantification of collagen deposition in lung sections from mice treated as in **c**. Total numbers of **j** neutrophils, **k** lymphocytes, and **l** leukocytes (Mock+PBS–SARS-

CoV-2 + PBS, *P* = 0.000078) in the bronchoalveolar lavage fluid (BALF) of mice treated as in **c**. Viral titers in the **m** BALF and in **n** lung homogenates from SARS-CoV-2-infected mice treated as in **c**, detected using plaque assays. *n* = 5, 6 per group for **i**–**n**. **o** Clinical scoring of mice (assessed for weight loss, activity level, eye closure, and appearance of fur and posture) treated as in **c**. Clinical scoring of mice assigned SARS-CoV-2-infected mice to both categories 2 and 3 (83% to category 2, 17% mice to category 3), while SARS-CoV-2-infected mice that received therapeutic administration of 4-OI were dispersed across no clinical score, category 1, and category 2 groups (33% mice to no clinical score, 50% mice to category 1, and 17% mice to category 2). Quantification of **p** *Ifnb1* (Mock+PBS–SARS-CoV-2 + PBS, *P* = 0.00003) and **q** *Nfe2l2* mRNA by qRT-PCR in PBMCs of mice treated as in **c** (*n* = 5, 6 per group). Human PBMCs from healthy controls (HC) (*n* = 5, 6) and SARS-CoV-2-infected patients (*n* = 9–11) were assessed for quantification of **r** *F3* and **s** *CASP4* mRNA by qRT-PCR. Data from **a, b** and **r, s** are mean ± SEM from three independent in vitro experiments. Data from **e, g, i**–**n**, and **p, q** are mean ± SD. Scale bar for **c, d, f**, and **h** = 50 µm. *P* values calculated using two-tailed Student's *t* test for paired comparisons or one-way ANOVA for multiple comparisons. Source data are provided as a Source Data file.

Collectively, our work supports recent clinical trials testing anti-inflammatory compounds as potential inhibitors of inflammation-associated coagulation[75–78]. There is an extensive crosstalk between inflammation and coagulation and therefore the broad reduction in inflammation exerted by DMF and 4-OI may contribute to their anticoagulant properties. However, our study identifies a targetable pathway (the type I IFN-TF axis) in thromboinflammation and our work supports redeploying the clinically approved DMF in clinical trials for inflammation-associated coagulopathies, in particular for autoimmune and thromboinflammatory diseases including systemic lupus erythematosus (SLE) and antiphospholipid syndrome (APS), which are characterized by excessive type I IFN production and are associated with significantly greater risk of developing thrombotic cardiovascular disease[84,85]. This is likely due in part to significantly elevated expression of *F3* mRNA in PBMCs from patients with SLE[86] and APS[87,88], which correlates with risk of vascular disease. Our work also supports the future development of 4-OI-based compounds as a new class of anticoagulants that inhibit inflammation-associated coagulation. For example, DMF and 4-OI-based compounds may have utility in sepsis-associated DIC, which is exacerbated by increased thrombin levels in patients[72].

In summary, these anticoagulant effects that we describe add to the well-characterized anti-inflammatory properties of DMF and 4-OI, further supporting their use in infectious diseases where both inflammation and coagulation are key pathologic features.

## Methods
### Animal details
All mice were on a C57BL/6 JOlaHsd background unless stated below. Wild-type (WT) mice were bred in-house. *Caspase-11*$^{-/-}$ mice on the C57BL/6 J background were backcrossed onto the C57BL/6 J background for another 8 generations. Heterozygous breeding pairs were used to generate wild-type and *Caspase-11*$^{-/-}$ littermates, which were used for all experiments described. *Ifnar*$^{-/-}$ mice were also generated on the C57BL/6J background and age- and sex-matched WT mice were used in all experiments described. Experiments were performed with 6-to-12-week-old male and female mice bred under specific pathogen-free conditions, under license and approval of the local animal research ethics committee (Health Products Regulatory Authority) and European Union regulations. In vivo models were performed with 6-12-week-old C57BL/6JOlaHsd mice and littermates were randomly assigned to experimental groups. All animal procedures were ethically approved by the Trinity College Dublin Animal Research Ethics Committee prior to experimentation, and conformed with the Directive 2010/63/EU of the European Parliament.

### Generation of murine BMDMs
6–12-week-old mice were euthanized in a $CO_2$ chamber and death was confirmed by cervical dislocation. Bone marrow was subsequently harvested from the tibia, femur, and ilium. For monocyte/macrophage isolation, cells were differentiated in DMEM containing L929 supernatant (10%), fetal calf serum (FCS) (10%), and penicillin/streptomycin (1%) for 6 days, after which cells were counted and plated at $0.5 \times 10^6$ cells/mL unless otherwise stated.

### Isolation of human PBMCs
Human blood samples from healthy donors and SARS-CoV-2-infected patients were collected and processed at the School of Biochemistry and Immunology in TBSI (TCD) or the St James's, Tallaght University Hospital, Trinity Alliance for Research (STTAR) Bioresource, St. James's Hospital, Tallaght University Hospital, and Trinity Translational Medicine Institute (TCD) (for PBMCs isolated from SARS-CoV-2-infected patients). Blood samples were obtained anonymously and written informed consent for the use of blood for research purposes was obtained from the donors. All the procedures involving experiments on human samples were approved by the School of Biochemistry and Immunology Research Ethics Committee (TCD). The collection of human samples for STTAR was approved by the National Research Ethics Committee. Experiments were conducted according to the TCD guide on good research practice, which follows the guidelines detailed in the National Institutes of Health Belmont Report (1978) and the Declaration of Helsinki. 30 mL whole blood was layered on 20 mL Lymphoprep (StemCell Technologies, Inc.), followed by centrifugation for 20 min at $400 \times g$ with the brake off, after which the upper plasma layer was removed and discarded. The layer of mononuclear cells at the plasma-density gradient medium interface was retained, and 20 mL PBS was added. Cells were centrifuged for 8 min at $300 \times g$ and the resulting supernatant was removed and discarded. The remaining pellet of mononuclear cells was resuspended, counted and plated at $1 \times 10^6$ cells/mL in RPMI supplemented with FCS (10%) and penicillin-streptomycin (1%).

### Reagents
Dimethyl fumarate (DMF) (242926) and 4-octyl itaconate (4-OI) (SML2338) were dissolved in dimethyl sulfoxide (DMSO) (D8418), all purchased from Sigma. Baricitinib (7222) was purchased from R&D and also dissolved in DMSO. Recombinant mouse IFN-β was purchased from BioLegend (581304). High molecular weight poly(I:C) (tlrl-pic) was purchased from Invivogen. LPS from *E. coli* (ALX-581-010-L002) was purchased from Enzo Life Sciences for in vitro experiments. LPS from *E. coli* (L4524) and heparin (SRE-0027) were purchased from Sigma for in vivo experiments. 1H1 anti-TF antibody was a kind gift

**Table 1 | Mouse and human primer sequences for quantitative real-time PCR**

| Mouse primers | |
| --- | --- |
| **Primer** | **Sequence (5′–3′)** |
| Casp4_F | GGTGGTGAAAGAGGAGCTTAC |
| Casp4_R | CCAGGAATGTGCTGTCTGAT |
| F3_F | ACCCAAACCCACCAACTATAC |
| F3_R | GGTCACATCCTTCACGATCTC |
| Gclm_F | TGGAGTTCCCAAATCAGCCC |
| Gclm_R | TGCATGGGACATGGTGCATT |
| Hmox1_F | CCTCACAGATGGCGTCACTT |
| Hmox1_R | GCTGATCTGGGGTTTCCCTC |
| Ifnb1_F | ATGGTGGTCCGAGCAGAGAT |
| Ifnb1_R | CCACCACTCATTCTGAGGCA |
| Isg15_F | GCCAGAAGCAGACTCCTTAAT |
| Isg15_R | ACACCAGGAAATCGTTACCC |
| Rps18_F | ATGTGAAGGATGGGAAGTACAG |
| Rps18_R | CCCTCTATGGGCTCGAATTT |
| Usp18_F | TCAGTGCCTGCAGAAATACA |
| Usp18_R | TCCGTGATCTGGTCCTTAGT |

| SARS-CoV-2 primers | |
| --- | --- |
| **Primer** | **Sequence (5′–3′)** |
| Hprt_F | AGGCCAGACTTTGTTGGATTTGAA |
| Hprt_R | CAACTTGCGCTCATCTTAGGCTTT |
| Ifnb1_F | AACTCCACCAGCAGACAGTG |
| Ifnb1_R | GGTACCTTTGCACCCTCCAG |
| Ifng_F | GAGGTCAACAACCCACAGGT |
| Ifng_R | GGGACAATCTCTTCCCCACC |
| Il1b_F | GAGGACATGAGCACCTTCTTT |
| Il1b_R | GCCTGTAGTGCAGTTGTCTAA |
| Nfe2l2_F | GGACATGGAGCAAGTTTGGC |
| Nfe2l2_R | CCAGCGAGGAGATCGATGAG |

| Human primers | |
| --- | --- |
| **Primer** | **Sequence (5′–3′)** |
| CASP4_F | GCTCTTCAACGCCACACAAC |
| CASP4_R | GGTGGGCATTTGAGCTTTGG |
| F3_F | CGCTTCAGGCACTACAAATAC |
| F3_R | GACTTGATTGACGGGTTTGG |
| RPS13_F | TCACCGTTTGGCTCGATATT |
| RPS13_R | GGCAGAGGCTGTAGATGATT |

from Dr Helen Bettencourt (Genentech, Inc. South San Francisco, CA)[46]. Cells were transfected (to activate caspase-11) using FuGENE HD (Promega).

**Non-canonical inflammasome assay**

Cells were treated with DMF (5–10 µM) and 4-OI (125–250 µM) for 1 h before being primed with LPS (100 ng/mL) for 3 h, after which the medium was replaced and 2 mg LPS was transfected using FuGENE HD (Promega) overnight (16 h) in serum-free media in order to activate the inflammasome by cleaving and activating caspase-11[19].

**RNA isolation and quantitative real-time PCR**

Total RNA was isolated using the PureLink RNA Mini Kit (Invitrogen) and quantified using a Nanodrop 2000 UV-visible spectrophotometer. cDNA was prepared using 20–100 ng/µL total RNA by a reverse transcription-polymerase chain reaction (RT-PCR) using a high-capacity cDNA reverse transcription kit (Applied Biosystems), according to the manufacturer's instructions. Real-time quantitative PCR (qPCR) was performed with an ABI 7500 Fast real-time PCR system (Applied Biosystem) on cDNA using SYBR Green (Invitrogen). Data were normalized to murine *Rps18* or *Hprt* and human *RPS13* as endogenous controls, and mRNA expression fold change relative to controls was calculated using the $2^{-\Delta\Delta Ct}$ method. All fold changes are expressed normalized to the untreated control. Table 1 lists the primer sequences that were used:

**RNA silencing (siRNA)**

Pre-designed mouse-specific silencer select siRNAs for *Jak1* (s68537; Ref Seq NM_146145; sense sequence: CACUGAUUGUCCACAAUAUTT; antisense sequence: AUAUUGUGGACAAUCAGUGGG), *Nrf2* (s70522; Ref Seq NM_010902; sense sequence: CAGGAGAGGUAAGAAUAAATT; antisense sequence: UUUAUUCUUACCUCUCCUGCG), and negative control (4390843) were ordered from Thermo Fisher. Cells were transfected with 50 nM siRNA using 5 µL lipofectamine RNAiMAX according to manufacturer's instructions (Thermo Fisher). Cells were transfected in medium without serum and antibiotics which was replaced with complete medium 8 h later. Cells were subsequently left for a further 12 h prior to treatment.

**RNA sequencing**

BMDMs (3 independent mice) were treated as indicated and RNA was extracted as previously detailed. mRNA was extracted from total RNA using poly-T-oligo-attached magnetic beads. After fragmentation, the first strand cDNA was synthesized using random hexamer primers, followed by the second strand cDNA synthesis. The library was checked with Qubit and RT-PCR for quantification and bioanalyzer for size distribution detection. Quantified libraries were pooled and sequenced on the NovaSeq 6000 S4 (Illumina).

**ELISA**

A DuoSet ELISA kit for IFN-β (R&D) in the supernatant of BMDMs was purchased and carried out according to the manufacturer's instructions with cell supernatants added to each plate in duplicate or triplicate. Absorbance was read at 450 nm and quantified using a FLUOstar Optima plate reader. Corrected absorbance values were calculated by subtracting the background absorbance and cytokine concentrations were subsequently obtained by extrapolation from a standard curve plotted on GraphPad Prism 9.0.

**Western blotting**

Supernatant was removed from cells following stimulation and lysates were harvested in 30–50 µL lysis buffer (0.125 M Tris pH 6.8, 10% glycerol, 0.02% SDS, 5% DTT). Lysates were subsequently heated to 95 °C for 5 min to denature proteins. To concentrate supernatants for western blot, 5 µL Strataclean Resin (Agilent) was added to 500 µL of supernatant and vortexed for 1 min. Supernatants were then centrifuged at 210 x g for 2 min at 4 °C. Supernatants were removed and discarded, and the remaining pellet was resuspended in 30 µL lysis buffer. SDS-PAGE was used to resolve proteins by molecular weight. Samples were boiled at 95 °C for 5 min prior to loading into a 5% stacking gel. The percentage resolving gel (8–12%) depended on the molecular weight of the given protein. The Bio-Rad gel running system was used to resolve proteins and the Bio-Rad wet transfer system was used for the electrophoretic transfer of proteins onto PVDF membrane. Following transfer, the membrane was incubated in blocking solution (5% milk powder or 5% BSA in TBST) for 1 h and subsequently incubated in primary antibody (5% milk powder or 5% BSA in TBST) rolling overnight at 4 °C. The membrane was incubated for 1 h with secondary antibody (diluted in 5% milk powder or 5% BSA in TBST) at room temperature. Prior to visualization, the membrane was immersed in WesternBright ECL Spray (Advansta). Proteins were

visualized on a ChemiDoc MPTM Imaging System (Bio-Rad), and both chemiluminescent and white light images were taken. Quantification of western blot images was performed using Image Lab Software (Bio-Rad). Adjusted band volume was calculated for each band and for each experimental condition this was presented as target protein/house-keeping protein. Uncropped and unprocessed blots are available as source data in the Source Data file.

The following antibodies were used for western blotting: mouse-reactive anti-rat CASPASE-11 (14340), anti-rabbit GAPDH (2118), anti-rabbit JAK1 (3344), anti-rabbit α-TUBULIN (2144), and anti-rabbit TF (44861) were purchased from Cell Signaling. Working dilutions of primary antibodies for western blotting were 1:1000. Horseradish peroxidase (HRP)-conjugated anti-rat (112-035-003) and anti-rabbit (111-035-003) IgG antibodies (both 1:2500) were purchased from Jackson Immunoresearch. All antibodies have been validated for mouse reactivity and for western blotting.

## LDH assay
The CytoTox 96 Non-Radioactive Cytotoxicity Assay (Promega) was used to quantify lactate dehydrogenase (LDH) release from cells as a measure of pyroptotic cell death in BMDMs following non-canonical inflammasome stimulation. Freshly harvested supernatants were used in this assay. 25 μL of each supernatant was added to 25 μL Cytotox 96 Reagent and incubated in the dark at room temperature for 30 min. 12.5 μL acetic acid was added to stop the reaction, and the absorbance at 492 nm was measured using a FLUOstar optima. 100 μL total lysis solution was added to untreated cells 30 min before harvesting and served as a maximum LDH release control. Medium alone was used to correct for background absorbance.

## Live-cell imaging
Pyroptosis was visualized by adding propidium iodide (PI) (R37169, Invitrogen) (1 drop/$1 \times 10^6$ cells/mL) at time of transfection and imaged using the IncuCyte S3 Live-Cell Analysis System (Essen BioScience). PI is not permeant to live cells; pyroptotic dead cells stained red. Images and movies were saved in tiff format, exported, and analyzed in ImageJ (NIH).

## LPS (endotoxin)-induced model of inflammation with associated coagulopathy
6-week-old female mice were used, and littermates were randomly assigned to experimental groups. Compounds were resuspended in 10% DMSO followed by 90% cyclodextrin/PBS (40% w/v). For therapeutic assessment, mice were challenged with intraperitoneally (i.p.) injected LPS (1 mg/kg) for 2 h before i.p. injection of DMF (50 mg/kg), 4-OI (50 mg/kg), baricitinib (50 mg/kg), 1H1 anti-TF antibody (15 mg/kg), or vehicle (PBS in 40% cyclodextrin) for 4 h. Mice were euthanized in a $CO_2$ chamber and lungs were harvested and whole blood samples were collected in 3.2% sodium citrate-coated tubes to prevent coagulation. For platelet-poor plasma (PPP) isolation, whole blood was spun at 2000 rpm at 4 °C for 15 min, and plasma (the top, liquid layer of the buffy coat) was used fresh. For the survival trial, mice were challenged i.p. with LPS (15 mg/kg) for 24 h before treatment i.p. with DMF (50 mg/kg), 4-OI (50 mg/kg), or vehicle (PBS in 40% cyclodextrin) for 48 h. Heparin (200 IU/kg) was injected subcutaneously 24 h after LPS injection. Mice were assessed every 6 h for 72 h post LPS challenge, and graded clinically from 0 to 3 (assessing weight loss, activity level, eye closure, and appearance of fur and posture), with 0 representing healthy and 3 reaching humane endpoint. Mice reaching humane endpoint were euthanized in a $CO_2$ chamber.

## Staphylococcus aureus-systemic infection model with coagulopathy
For in vitro infection with S. aureus, BMDMs were treated with DMF (5–10 μM) or 4-OI (125–250 μM) for 1 h before being primed with $5 \times 10^6$ CFU LAC for 3 h prior to harvesting cell lysates. For the community-acquired MRSA strain S. aureus USA300-LAC in vivo, mice were co-treated via intravenous injection with $5 \times 10^7$ CFU LAC ± DMF (50 mg/kg), 4-OI (50 mg/kg), or PBS in 40% cyclodextrin for 6 h before harvesting as per our LPS (endotoxin)-induced septic shock model. For S. aureus PS80-strain, mice were co-treated i.p. with $5 \times 10^8$ CFU PS80 ± DMF (50 mg/kg), 4-OI (50 mg/kg), or PBS in 40% cyclodextrin for 6 h before a second dose of inhibitor and harvesting after 18 h.

## E. coli-induced systemic infection model with coagulopathy
Mice were co-treated i.p. with $1 \times 10^7$ CFU CFT073 E. coli ± DMF (50 mg/kg), 4-OI (50 mg/kg), or PBS in 40% cyclodextrin for 6 h before a second dose of inhibitor and harvesting after 18 h.

## SARS-CoV-2 model of inflammation with associated coagulopathy
Heterozygous male K18-hACE c57BL/6 J mice (strain: 2B6.Cg-Tg(K18-ACE2)2Prlmn/J) were obtained from The Jackson Laboratory. Animals were housed in groups and fed standard chow diets. Mice were administered $10^3$ PFU SARS-CoV-2 (Wuhan isolate; VIC01/2020) via intranasal administration and intranasally treated with 4-OI (10 mg/kg) from 1 day post infection, with daily treatments continuing until the termination of the experiment at day 6 post infection. Numbers of total neutrophils, lymphocytes, and leukocytes were assessed in the bronchoalveolar lavage fluid (BALF) and lung homogenates of mice. A plaque assay was utilized to assess viral titers from SARS-CoV-2-infected mice. Collagen deposition was quantified using Sirius Red-stained lung histological sections, with the area surrounding the primary airways (red) enumerated for each mouse. PBMCs were isolated from whole blood for assessment of mRNA expression. Clinical scoring of mice assessed weight loss, activity level, eye closure, and appearance of fur and posture. Mice were graded clinically with increasing severity in four categories, from 0 to 3.

## Cell-based thrombin generation assay
Cells were seeded onto a 96-well plate and left overnight to adhere. Following the described cell stimulations, supernatants were removed and cells were washed once with PBS. MP-reagent (Thrombinoscope) was used as a source of phospholipids in 80 μL FXII-deficient plasma (Haemtech). Thrombin generation of cells in situ in the plate was measured by comparing experimental wells against thrombin activity of a calibrator, as recently described[39]. As BMDM TGA was carried out with cells in situ, changes in thrombin generation will most likely be as a result of surface-bound TF rather than TF released from the surface. The reaction was initiated with 20 μL FluCa-kit (0.42 mM fluorometric substrate, 16.67 nM $CaCl_2$). Fluorescence was quantified using Thrombinoscope software on the Fluoroskan for 60 min. Data was analyzed on GraphPad Prism 9.0.

## Plasma thrombin generation assay
Plasma was diluted 1:3 for all experiments described. MP-reagent which contains phospholipids was added to wells to allow coagulation complex formation. Mouse thrombin converts the fluorogenic substrate at a rate which is 20% lower than human thrombin, therefore to correct for this discrepancy, calibrator activity was set to 20% higher in the Thrombinoscope settings. The temperature of the Fluoroskan was set to 33 °C. 20 μL plasma and 20 μL of MP-reagent were added to wells and the reaction was initiated with 20 μL FluCa-kit (0.42 mM fluorometric substrate, 8.2 nM $CaCl_2$). Fluorescence was quantified using Thrombinoscope software on the Fluoroskan for 60 min. Data was analyzed on GraphPad Prism 9.0.

## Histology
The superior, middle, and inferior lobes of right lungs were rapidly removed from mice, fixed with 4% paraformaldehyde in PBS, and

embedded in paraffin using an automatic tissue processor (Leica Microsystems). 3 µM-thick lung sections were deparaffinized in xylene and rehydrated through graded ethanol washes. The antigen retrieval was performed in Tris-EDTA buffer (10 mM Tris pH 9.0, 1 mM EDTA) for 30 min at 95 °C followed by incubation with proteinase K for 5 min at room temperature. The lung tissue sections were blocked with 10% BSA in TBS for 1 h at room temperature and then incubated overnight at 4 °C with either a rabbit anti-fibrinogen/fibrin (cat. no.: A 0080; Dako, Gostrup, Denmark), a rabbit anti-von Willebrand factor (VWF; cat. no.: A 0082; Dako), or a rabbit anti-Tissue Factor antibody (Novus Bio; NBP2-15139). Afterwards, the lung tissue sections were extensively washed with TBS and incubated with a secondary antibody labeled with Alexa Fluor™ 555 (Thermo-Fisher Scientific; Waltham, MA) for 1 h at room temperature. Finally, the slides were embedded in Vectashield Mounting Medium with DAPI (Vector Laboratories Inc., Burlingame, CA). Images were taken with a Leica DM 6000 microscope (Leica, Wetzlar, Germany) using a 20x objective and processed using Leica Application Suite Advanced Fluorescence (LAS AF) software. Alternatively, antigen detection was performed using a Zytochem-Plus AP Polymer Kit in accordance with the manufacturer's instructions (Zymed Systems, Berlin, Germany) and the slides were scanned with a Miramax slide scanner (Zeiss, Oberkochen Germany).

## Quantification, statistical analysis, and reproducibility

Details of all statistical analyses performed can be found in the figure legends. Data were expressed as mean ± standard error of the mean (SEM) unless stated otherwise. $P$ values were calculated using two-tailed Student's $t$ test for pairwise comparison of variables and one-way ANOVA for multiple comparison of variables. A Sidak's or Tukey's multiple comparisons test was used as a post-test when performing an ANOVA. A Mantel–Cox test was used for log-rank analysis of the survival Kaplan–Meier curve. A confidence interval of 95% was used for all statistical tests. For statistical significance, exact $P$ values are included with each panel, and for $P$ values < 0.0001, exact $P$ values are included in the figure legends. Sample sizes were determined on the basis of previous experiments using similar methodologies. All depicted data points are biological replicates taken from distinct samples. Each figure consists of a minimum of three independent experiments from multiple biological replicates unless stated otherwise in the figure legends. $n$ = the number of animals or the number of independent experiments with primary BMDMs. For in vivo studies, age- and sex-matched mice were randomly assigned to treatment groups, unless otherwise stated.

## Reporting summary

Further information on research design is available in the Nature Portfolio Reporting Summary linked to this article.

## Data availability

RNA sequencing data (https://doi.org/10.5061/dryad.6wwpzgn28) is available via the Dryad Data Platform. Data generated from the Immunological Genome Project (ImmGen) bulk-population RNA-seq database[38], Interferome v2.0[42], and the ChIP-Atlas enrichment analysis tool[43] are publicly available. Source data are provided with this paper in the Supplementary Information. Source data are provided with this paper.

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

## Acknowledgements

We would like to thank the O'Neill laboratory for helpful discussions, in particular Dr. Anne F. McGettrick and Ms. Kathy Banahan for assistance in the laboratory and with PBMC isolation. We also thank Dr. Helen Bettencourt (Genentech, Inc., South San Francisco, CA) for the kind gift of the 1H1 anti-TF antibody. We are especially grateful to the patients and staff that contributed samples and data to the STTAR Bioresource (St. James's Hospital, Dublin) for the assessment of COVID-19 markers in human PBMCs. We also thank Ms. Nicole Wood and the Wellcome Trust Clinical Research Facility (St. James's Hospital, Dublin) for their support. L.A.J.O'N. acknowledges grant support from the European Research Council Metabinnate (834370), the Wellcome Trust (205455), and Science Foundation Ireland (19/FFP/6507). J.S.O'D. acknowledges grant support from Science Foundation Ireland (20/FFP-A/8952). R.M.M. acknowledges grant support from the Wellcome Trust (202846/Z/16/Z) and Science Foundation Ireland (15/IA/3041). M.W. acknowledges grant support from the German Research Foundation (DFG: SFB/TR84699 Project A2) and the German Center for Lung Research (82 DZL 005A1).

## Author contributions

T.A.J.R., Z.Z., and L.A.J.O'N. conceptualized the project. T.A.J.R. was the lead experimentalist and designed and performed in vitro and in vivo experiments, analyzed and interpreted data, and wrote the manuscript. A.H. conducted in vitro experiments, analyzed and interpreted data, assisted with specimen monitoring, and provided critical input. A.M.R. conducted in vitro experiments, performed thrombin generation assays, and analyzed and interpreted data. M.D.J. performed in vivo experiments and analyzed and interpreted data. E.C.O'B. performed in vivo experiments. J.E.T-K., M.M.W., E.A.D., S.M., and N.G.H. assisted with in vivo experiments. H.J.W., C.G.P., and A.Z. assisted with specimen monitoring. P.S., F.S., and C.N. performed histological analysis. E.G.V. performed in vitro experiments. L.O'D., S.G., A.L., J.D., C.N.C, and N.C. provided PBMCs from COVID-19-infected patients. M.C., P.G.F., and K.H.G.M. provided helpful discussions. G.M. advised on live-cell imaging and assisted with the preparation of samples for histological analysis. E.M.C. supplied Caspase-11$^{-/-}$ mice. J.S.O'D., P.M.H., and R.M.M. provided critical input. P.J.H. assisted with and advised on transcription factor promoter analyses. M.W. performed and advised on histological analysis. R.J.S.P. assisted with experimental design and provided critical input. Z.Z. designed experiments and oversaw a portion of the research project. L.A.J.O'N. obtained funding and oversaw the research project.

## Competing interests

The authors declare no competing interests.

## Additional information

Tristram A. J. Ryan ⓘ[1], Alexander Hooftman ⓘ[1], Aisling M. Rehill ⓘ[2], Matt D. Johansen ⓘ[3], Eóin C. O' Brien[4], Juliana E. Toller-Kawahisa[1], Mieszko M. Wilk ⓘ[1,5], Emily A. Day[1], Hauke J. Weiss[1], Pourya Sarvari[6], Emilio G. Vozza[4], Fabian Schramm[6], Christian G. Peace ⓘ[1], Alessia Zotta[1], Stefan Miemczyk[3], Christina Nalkurthi[3], Nicole G. Hansbro[3], Gavin McManus[1], Laura O'Doherty ⓘ[7,8,9], Siobhan Gargan[9], Aideen Long[9], Jean Dunne[10], Clíona Ní Cheallaigh[7,9], Niall Conlon[8,9,10], Michael Carty[1], Padraic G. Fallon ⓘ[9,11], Kingston H. G. Mills[1], Emma M. Creagh ⓘ[1], James S. O' Donnell[2], Paul J. Hertzog[12,13], Philip M. Hansbro ⓘ[3], Rachel M. McLoughlin[4], Małgorzata Wygrecka ⓘ[6], Roger J. S. Preston[2], Zbigniew Zasłona[1] & Luke A. J. O'Neill ⓘ[1] ✉

[1]School of Biochemistry and Immunology, Trinity Biomedical Sciences Institute, Trinity College Dublin, Dublin 2, Ireland. [2]Irish Centre for Vascular Biology, School of Pharmacy and Biomolecular Sciences, RCSI University of Medicine and Health Sciences, Dublin 2, Ireland. [3]Centre for Inflammation, Centenary Institute and University of Technology Sydney, Faculty of Science, Sydney, NSW, Australia. [4]Host Pathogen Interactions Group, School of Biochemistry and Immunology, Trinity Biomedical Sciences Institute, Trinity College Dublin, Dublin 2, Ireland. [5]Department of Immunology, Faculty of Biochemistry, Biophysics and Biotechnology, Jagiellonian University, Kraków, Poland. [6]Center for Infection and Genomics of the Lung, German Center for Lung Research (DZL), Faculty of Medicine, Justus Liebig University, Giessen, Germany. [7]Department of Infectious Diseases, St. James's Hospital, Dublin, Ireland. [8]Clinical Research Facility, St. James's Hospital, Dublin, Ireland. [9]Department of Clinical Medicine, School of Medicine, Trinity Translational Medicine Institute, Trinity College Dublin, Dublin, Ireland. [10]Department of Immunology, St James's Hospital, Dublin, Ireland. [11]School of Medicine, Trinity Biomedical Sciences Institute, Trinity College, Dublin 2, Ireland. [12]Centre for Innate Immunity and Infectious Diseases, Hudson Institute of Medical Research, Clayton, VIC, Australia. [13]Department of Molecular and Translational Science, Monash University, Clayton, VIC, Australia. ✉e-mail: laoneill@tcd.ie

