## [Peer Review File · Nature Communications]

Dimethyl fumarate and 4-octyl itaconate are anticoagulants that suppress Tissue Factor in macrophages via inhibition of Type I InterferonThis manuscript has been previously reviewed at another journal that is not operating a transparent peer review scheme. This document only contains reviewer comments and rebuttal letters for versions considered at *Nature Communications*.

REVIEWERS' COMMENTS

Reviewer #2 (Remarks to the Author):

The authors provide compelling new data that addresses many of the concerns initially raised in my review. The new experiments bolster the model proposed and make a compelling story. I have no further issues.

Reviewer #3 (Remarks to the Author):

Dr Ryan and colleagues report in this paper on the impact of dimethyl fumarate (DMF) and an Itaconate derivate (4-OI) on the tissue factor dependent (TF) activation of the coagulation system. The main findings of this study are i) DMF and 4-OI inhibit TF-mediated coagulopathy via inhibition of the macrophage type I IFN-TF axis. The authors are to be congratulated for an extensive work that has been done for the revision of this manuscript. I enjoyed reading your report. The authors addressed very elegantly all of my comments.

Reviewer #4 (Remarks to the Author):

Ryan and colleagues describe how the type I interferon (IFN-I)-TF axis in macrophages can be modulated by DMF and 4-OI. They also prove that DMF and 4-OI inhibition of IFN-I also prevents caspase-11 induction. The authors use several mouse models of inflammation and infection and convincingly show that utilization of DMF and 4-OI is an effective intervention to decrease thromboinflammation.

The authors have been responsive to the original critiques and strongly improved their manuscript. There is only one point that in my opinion may need to be further amended or better discussed.

As also reported in previous reviews, it is hard to understand whether the anti-inflammatory activity of DMF and 4-OI is solely, or mostly, due to the inhibition of IFN-I (and its consequences on TF and caspase-11) or whether there is a general inhibition of the immune response due to the block of other inflammatory pathways.

Although the authors show that NRF2 is activated, this may be expected due to the fact that both DMF and 4OI activate NRF2. I would thus suggest to: i) if the authors have the data, show other pro- or anti-inflammatory cytokine levels for their in vitro and in vivo (as they already did in Sup Fig. 6a,b for the CoV2 infection) experiments; ii) if this is not possible in the short term, extensively discuss and highlight in their discussion that the anti-inflammatory effects in response to DMF and 4OI may also be due to other activities beside inhibition of the IFN-I-TF/Caspase-11 pathway.

Finally, Figure 3G reports Caspase-11KO mice, but the text specifies that an inhibitor has been utilized. The authors may want to consolidate this discrepancy.

Dimethyl fumarate and 4-octyl itaconate are anticoagulants that suppress Tissue Factor in macrophages via inhibition of Type I Interferon

Tristram A.J. Ryan¹, Alexander Hooftman¹, Aisling M. Rehill², Matt D. Johansen³, Eóin C. O'Brien⁴, Juliana E. Toller-Kawahisa¹, Mieszko M. Wilk^{1,5}, Emily A. Day¹, Hauke J. Weiss¹, Pourya Sarvari⁶, Emilio G. Vozza⁴, Fabian Schramm⁶, Christian G. Peace¹, Alessia Zotta¹, Stefan Miemczyk³, Christina Nalkurthi³, Nicole G. Hansbro³, Gavin McManus¹, Laura O'Doherty^{7,8,9}, Siobhan Gargan⁹, Aideen Long⁹, Jean Dunne¹⁰, Clíona Ní Cheallaigh^{7,9}, Niall Conlon^{8,9,10}, Michael Carty¹, Padraic G. Fallon^{9,11}, Kingston H. G. Mills¹, Emma M. Creagh¹, James S. O'Donnell², Paul J. Hertzog^{12,13}, Philip M. Hansbro³, Rachel M. McLoughlin⁴, Małgorzata Wygrecka⁶, Roger J.S. Preston², Zbigniew Zastona¹, Luke A.J. O'Neill^{1,14}

¹ School of Biochemistry and Immunology, Trinity Biomedical Sciences Institute, Trinity College Dublin, Dublin 2, Ireland

² Irish Centre for Vascular Biology, School of Pharmacy and Biomolecular Sciences, RCSI University of Medicine and Health Sciences, Dublin 2, Ireland

³ Centre for Inflammation, Centenary Institute and University of Technology Sydney, Faculty of Science, Sydney, NSW, Australia

⁴ Host Pathogen Interactions Group, School of Biochemistry and Immunology, Trinity Biomedical Sciences Institute, Trinity College Dublin, Dublin 2, Ireland

⁵ Department of Immunology, Faculty of Biochemistry, Biophysics and Biotechnology, Jagiellonian University, Kraków, Poland

⁶ Center for Infection and Genomics of the Lung, German Center for Lung Research (DZL), Faculty of Medicine, Justus Liebig University, Giessen, Germany

⁷ Department of Infectious Diseases, St. James's Hospital, Dublin, Ireland

⁸ Clinical Research Facility, St. James's Hospital, Dublin, Ireland

⁹ Department of Clinical Medicine, School of Medicine, Trinity Translational Medicine Institute, Trinity College Dublin, Dublin, Ireland

¹⁰ Department of Immunology, St James's Hospital, Dublin, Ireland

¹¹ School of Medicine, Trinity Biomedical Sciences Institute, Trinity College, Dublin 2, Ireland

¹² Centre for Innate Immunity and Infectious Diseases, Hudson Institute of Medical Research, Clayton, VIC, Australia

¹³ Department of Molecular and Translational Science, Monash University, Clayton, VIC, Australia

¹⁴ Lead contact & correspondence: laoneill@tcd.ie

Manuscript **NCOMMS-23-03624-T**

Point-by-point response to reviewers' comments for final revisions

Referee #2 – Remarks to the Author:

The authors provide compelling new data that addresses many of the concerns initially raised in my review. The new experiments bolster the model proposed and make a compelling story. I have no further issues.

Referee #3 – Remarks to the Author:

Dr Ryan and colleagues report in this paper on the impact of dimethyl fumarate (DMF) and an Itaconate derivate (4-OI) on the tissue factor dependent (TF) activation of the coagulation system. The main findings of this study are i) DMF and 4-OI inhibit TF-mediated coagulopathy via inhibition of the macrophage type I IFN-TF axis. The authors are to be congratulated for an extensive work that has been done for the revision of this manuscript. I enjoyed reading your report. The authors addressed very elegantly all of my comments.

Referee #4 – Remarks to the Author:

Ryan and colleagues describe how the type I interferon (IFN-I)-TF axis in macrophages can be modulated by DMF and 4-OI. They also prove that DMF and 4-OI inhibition of IFN-I also prevents caspase-11 induction. The authors use several mouse models of inflammation and infection and convincingly show that utilization of DMF and 4-OI is an effective intervention to decrease thromboinflammation.

The authors have been responsive to the original critiques and strongly improved their manuscript. There is only one point that in my opinion may need to be further amended or better discussed.

As also reported in previous reviews, it is hard to understand whether the anti-inflammatory activity of DMF and 4-OI is solely, or mostly, due to the inhibition of IFN-I (and its consequences on TF and caspase-11) or whether there is a general inhibition of the immune response due to the block of other inflammatory pathways.

Although the authors show that NRF2 is activated, this may be expected due to the fact that both DMF and 4OI activate NRF2. I would thus suggest to: i) if the authors have the data, show other pro- or anti-inflammatory cytokine levels for their in vitro and in vivo (as they already did in Sup Fig. 6a,b for the CoV2 infection) experiments; ii) if this is not possible in the short term, extensively discuss and highlight in their discussion that the anti-inflammatory effects in response to DMF and 4OI may also be due to other activities beside inhibition of the IFN-I-TF/Caspase-11 pathway.

Finally, Figure 3G reports Caspase-11KO mice, but the text specifies that an inhibitor has been utilized. The authors may want to consolidate this discrepancy.

We wish to record our appreciation to the reviewers and express our gratitude for their time and suggestions which have enhanced our manuscript. In addition, we thank the editor for their guidance and contributions throughout this peer review process. Building on the feedback from the reviewers, we hope we have now addressed all of their comments, for which we provide a point-by-point response below.

Point-by-point author response to comments from Referee #2:

The authors provide compelling new data that addresses many of the concerns initially raised in my review. The new experiments bolster the model proposed and make a compelling story. I have no further issues.

We thank reviewer #2 for their helpful comments which we believe have contributed to a stronger and more streamlined manuscript. We are pleased that we have successfully addressed many of the previous comments and that the reviewer found our new data to be compelling and worthy of publication.

Point-by-point author response to comments from Referee #3:

Dr Ryan and colleagues report in this paper on the impact of dimethyl fumarate (DMF) and an Itaconate derivate (4-OI) on the tissue factor dependent (TF) activation of the coagulation system. The main findings of this study are i) DMF and 4-OI inhibit TF-mediated coagulopathy via inhibition of the macrophage type I IFN-TF axis. The authors are to be congratulated for an extensive work that has been done for the revision of this manuscript. I enjoyed reading your report. The authors addressed very elegantly all of my comments.

We are grateful to the reviewer and their guidance with clarifying the focus on thromboinflammation within our manuscript. We are glad that the reviewer appreciates our extensive revisions and we are particularly pleased that they found our revised manuscript both enjoyable and elegant.

Point-by-point author response to comments from Referee #4:

Ryan and colleagues describe how the type I interferon (IFN-I)-TF axis in macrophages can be modulated by DMF and 4-OI. They also prove that DMF and 4-OI inhibition of IFN-I also prevents caspase-11 induction. The authors use several mouse models of inflammation and infection and convincingly show that utilization of DMF and 4-OI is an effective intervention to decrease thromboinflammation.

The authors have been responsive to the original critiques and strongly improved their manuscript. There is only one point that in my opinion may need to be further amended or better discussed.

We thank reviewer #4 for their positive feedback on our manuscript and the convincing nature of our results across several mouse models. We also appreciate that they found we had addressed the original comments which had strongly benefited our work.

As also reported in previous reviews, it is hard to understand whether the anti-inflammatory activity of DMF and 4-OI is solely, or mostly, due to the inhibition of IFN-I (and its consequences on TF and

caspase-11) or whether there is a general inhibition of the immune response due to the block of other inflammatory pathways.

Although the authors show that NRF2 is activated, this may be expected due to the fact that both DMF and 4OI activate NRF2. I would thus suggest to: i) if the authors have the data, show other pro- or anti-inflammatory cytokine levels for their in vitro and in vivo (as they already did in Sup Fig. 6a,b for the CoV2 infection) experiments; ii) if this is not possible in the short term, extensively discuss and highlight in their discussion that the anti-inflammatory effects in response to DMF and 4OI may also be due to other activities beside inhibition of the IFN-I-TF/Caspase-11 pathway.

We agree with the reviewer on this point and thank them for highlighting this. To acknowledge this point, we have amended the penultimate paragraph of our discussion to include the lines (new additions underlined):

“Collectively, our work supports recent clinical trials testing anti-inflammatory compounds as potential inhibitors of inflammation-associated coagulation. There is an extensive crosstalk between inflammation and coagulation and therefore the broad reduction in inflammation exerted by DMF and 4-OI may contribute to their anticoagulant properties. However, our study identifies a new, targetable pathway (the type I IFN-TF axis) in thromboinflammation and our work supports redeploying the clinically approved DMF in clinical trials for inflammation-associated coagulopathies, in particular for autoimmune and thromboinflammatory diseases including systemic lupus erythematosus (SLE) and antiphospholipid syndrome (APS), which are characterized by excessive type I IFN production and are associated with significantly greater risk of developing thrombotic cardiovascular disease”.

We hope with these additions that we have suitably addressed the reviewer’s helpful comment.

Finally, Figure 3G reports Caspase-11KO mice, but the text specifies that an inhibitor has been utilized. The authors may want to consolidate this discrepancy.

We appreciate this observation. The reviewer is indeed correct that we used *Casp11^{-/-}* knockout mice in this experiment. We also employed the antithrombin activator heparin as a control (and clinically approved) anticoagulant. It has recently been shown that heparin can also block caspase-11-mediated pyroptosis from macrophages as a key contributing factor to its anticoagulant properties (PMID 33561388), a paper which we have cited to clarify this point. In this way, we employed a genetic knockout as well as a clinically approved caspase-11 inhibitor to demonstrate its core role in inflammation-associated coagulation *in vivo*.